# Beneficial Impact of Eicosapentaenoic Acid on the Adverse Effects Induced by Palmitate and Hyperglycemia on Healthy Rat Chondrocyte

**DOI:** 10.3390/ijms25031810

**Published:** 2024-02-02

**Authors:** Chaohua Deng, Nathalie Presle, Anne Pizard, Cécile Guillaume, Arnaud Bianchi, Hervé Kempf

**Affiliations:** 1UMR 7365 CNRS-Université de Lorraine, Ingénierie Moléculaire et Physiopathologie Articulaire (IMoPA), Biopôle de l’Université de Lorraine, 54500 Vandoeuvre-les-Nancy, France; dengch@hust.edu.cn (C.D.); nathalie.presle@univ-lorraine.fr (N.P.); cecile.guillaume@univ-lorraine.fr (C.G.); herve.kempf@inserm.fr (H.K.); 2Department of Ophthalmology, Tongji Hospital, Tongji Medical College, Huazhong University of Science and Technology, Wuhan 430030, China; 3INSERM U955, Institut Mondor de Recherche Biomédicale (IMRB), Université Paris-Est-Créteil (UPEC), 94010 Créteil, France; anne.pizard@inserm.fr

**Keywords:** free fatty acid, chondrocyte, rat, EPA, osteoarthritis, hyperglycemia

## Abstract

Osteoarthritis (OA) is the most prevalent form of arthritis and a major cause of pain and disability. The pathology of OA involves the whole joint in an inflammatory and degenerative process, especially in articular cartilage. OA may be divided into distinguishable phenotypes including one associated with the metabolic syndrome (MetS) of which dyslipidemia and hyperglycemia have been individually linked to OA. Since their combined role in OA pathogenesis remains to be elucidated, we investigated the chondrocyte response to these metabolic stresses, and determined whether a n-3 polyunsaturated fatty acid (PUFA), i.e., eicosapentaenoic acid (EPA), may preserve chondrocyte functions. Rat chondrocytes were cultured with palmitic acid (PA) and/or EPA in normal or high glucose conditions. The expression of genes encoding proteins found in cartilage matrix (type 2 collagen and aggrecan) or involved in degenerative (metalloproteinases, MMPs) or in inflammatory (cyclooxygenase-2, COX-2 and microsomal prostaglandin E synthase, mPGES) processes was analyzed by qPCR. Prostaglandin E_2_ (PGE_2_) release was also evaluated by an enzyme-linked immunosorbent assay. Our data indicated that PA dose-dependently up-regulated the mRNA expression of *MMP-3* and *-13*. PA also induced the expression of *COX-2* and *mPGES* and promoted the synthesis of PGE_2_. Glucose at high concentrations further increased the chondrocyte response to PA. Interestingly, EPA suppressed the inflammatory effects of PA and glucose, and strongly reduced MMP-13 expression. Among the free fatty acid receptors (FFARs), FFAR4 partly mediated the EPA effects and the activation of FFAR1 markedly reduced the inflammatory effects of PA in high glucose conditions. Our findings demonstrate that dyslipidemia associated with hyperglycemia may contribute to OA pathogenesis and explains why an excess of saturated fatty acids and a low level in n-3 PUFAs may disrupt cartilage homeostasis.

## 1. Introduction

Osteoarthritis (OA) is the most common joint disease accounting for 50% of degenerative illnesses of the musculoskeletal system [1]. The pathology of OA involves the whole joint in a degenerative process that includes focal and progressive cartilage loss from fibrillations in the early stages of the disease to full-thickness matrix depletion. OA changes also include the development of osteophytes at the bone margins and the inflammation of the synovium. In normal conditions, the chondrocyte, the unique cell type residing in the adult cartilage matrix, is responsible for remodeling and maintaining the structural and functional integrity of the extracellular matrix. During OA, chondrocytes exhibit an aberrant behavior with increased matrix degradation and reduced anabolic processes, which results in an imbalance in cartilage homeostasis [2].

Various clinical OA phenotypes result from the pathological context in which OA arises. Among them, metabolic OA becomes the most prevalent as it is associated with metabolic syndrome (MetS) [3,4,5], which is now a major worldwide health issue defined by five core features, i.e., hypertension, dyslipidemia, visceral obesity, hyperglycemia and insulin resistance. Patients with MetS develop clinical symptoms of OA earlier in life and exhibit increased inflammation and pain in the joints, in comparison with OA patients in the absence of MetS [6].

If clinical conditions clustered in MetS may be individually linked to OA, the nature of the interaction between MetS and OA is complex and has not yet been fully evaluated. The increased joint loading resulting from obesity plays a role in OA development. However, this altered biomechanics cannot explain the association between obesity and hand OA [7,8]. The systemic release of pro-inflammatory cytokines by visceral adipose tissue in obese individuals leads to a chronic low-grade inflammatory state, which may induce damage in many peripheral tissues including joint tissues.

Other systemic metabolic disorders such as hyperglycemia and dyslipidemia may also contribute to OA pathogenesis. Epidemiological studies provide conflicting conclusions regarding the link between hyperglycemia and OA [9,10,11]. However, both in vitro and in vivo experiments have shown that a high glucose concentration may induce deleterious effects on cartilage through oxidative stress and advanced glycation end-products, which lead to both cellular and tissue changes in the joint [12,13].

Many studies have investigated the influence of both saturated and polyunsaturated fatty acids (PUFAs) on joint health [14]. The synovial fluid contains various free fatty acids with different profiles between OA and non-symptomatic control knee joints [15], the lipidomic profile being associated with OA severity [16,17]. As found in serum from obese individuals, the levels of total fatty acids increase in osteoarthritic cartilage specimens, especially in the superficial area of the OA cartilage [18]. Interestingly, these lipid droplets are associated with an increasing grade of lesion severity [19] and are found at the early stages of the disease before histological changes become apparent, suggesting that lipid deposition in the joint may initiate OA development [20]. Despite these observations, the contribution of fatty acids to joint diseases is rather complex and remains unclear as their deleterious or beneficial effects depend upon their biochemical structure. A high-fat diet with saturated fatty acids or n-6 PUFAs has been shown to be associated with synovitis, cartilage degradation and progression of OA in experimental models [21,22] and with more severe radiographic knee changes in OA patients [23]. By contrast, dietary n-3 PUFAs, also called omega-3, and which include eicosapentaenoic acid (EPA) and docosahexaenoic (DHA), reduce spontaneous OA in guinea pigs [24]. Similarly, an elevated level of endogenous n-3 PUFAs in fat-1 transgenic mice, which are able to endogenously convert n-6 PUFAs to n-3 PUFAs, alleviate articular cartilage destruction and osteophyte formation in the medial meniscus resection model of OA [25]. In fact, according to their carbon chain length, free fatty acids regulate various cellular processes and physiological functions via several G-protein-coupled receptors called free fatty acid receptors (FFARs). The long-chain saturated and unsaturated fatty acids function as signaling molecules through binding to FFAR1 (GPR40) and FFAR4 (GPR120), while short-chain fatty acids activate FFAR3 (GPR41) and FFAR2 (GPR43) [26].

Although collectively these data from clinical and experimental studies provide evidence for an individual role of hyperglycemia or dyslipidemia in OA pathogenesis, the combined effects of these two components of MetS remain to be elucidated. Therefore, qPCR and ELISA analysis were used to determine whether hyperglycemia together with palmitic acid (PA), one of the most prominent saturated fatty acids found in synovial fluid [27], alter rat chondrocyte functions, and whether EPA may modulate these effects. Lastly, with the help of pharmacological agonists and antagonists, we have sought to identify which of the FFARs is involved in rat chondrocyte response to PA associated with hyperglycemia and EPA.

## 2. Results

### 2.1. Rat Chondrocyte Response to PA

The expression of genes encoding components of the extracellular matrix, i.e., type 2 collagen and aggrecan, as well as matrix metalloproteinases-3 and -13 (MMP-3 and -13), which play a major role in cartilage degeneration, was used to evaluate the effects of PA on rat articular chondrocytes.

Increasing concentrations of the fatty acid ranging from 0.1 to 0.5 mM induced the expression of *MMP-3* and -*13* in a dose-dependent manner (Figure 1A,B). By contrast, the gene encoding type 2 collagen was not identified as a significant target of PA (Figure 1C). A trend toward a reduced expression of *aggrecan* was found in PA-treated cells, but without statistical significance (Figure 1D).

PA also dose-dependently stimulated the expression of two inflammatory enzymes involved in prostaglandin E_2_ (PGE_2_) synthesis, namely the *cyclooxygenase (COX)-2* and the *microsomal prostaglandin E synthase (mPGES)*. A more than 20-fold increase was achieved with the highest dose of PA (Figure 2A,B). Accordingly, this marked overexpression of COX-2 and mPGES led to a release of PGE_2_ in the culture medium of chondrocytes exposed to the fatty acid.

### 2.2. Influence of Hyperglycemia on PA Effects

The expression of *MMPs*, *type 2 collagen*, *aggrecan*, *COX-2* and *mPGES* remained unchanged upon treatment of articular cells with glucose at 4.5 g/L only, i.e., without PA (Figure 3). However, hyperglycemia potentiated the effects of PA since chondrocyte responses to a low dose of PA associated with a high level of glucose were becoming closer to the levels of regulation upon higher doses of PA under normoglycemic conditions. Chondrocytes exposed to PA (0.1 mM) together with glucose at 4.5 g/L exhibited an elevated expression of *MMP-13*, *COX-2* and *mPGES* compared to PA-treated cells in normoglycemic conditions (1 g/L). However, the difference between both groups reached statistical significance only for *COX-2* (*p* = 0.008). Moreover, the addition of a high glucose level to PA at 0.1 mM further supported the trend of PA to decrease the mRNA level of aggrecan previously found under normoglycemic conditions (*p* = 0.015 between PA-treated cells under normo- and hyperglycemia). By contrast, the effects of PA on the mRNA expression of gene encoding type 2 collagen and MMP-3 did not change under hyperglycemia.

### 2.3. Effects of EPA on Chondrocyte Response to Both PA and Hyperglycemia

Increasing concentrations of EPA (10, 30, 60 and 100 µM) were used to evaluate its influence on the inflammatory and degenerative effects of PA (0.1 mM) in high glucose conditions. EPA (100 µM) alone was not able to modulate gene expression (Figure 4). The n3-PUFA decreased the stimulatory effect of PA and glucose on *MMP-3* expression. However, the difference with chondrocytes treated with PA and high glucose did not reach statistical significance (*p* = 0.136, *p* = 0.07, *p* = 0.141 and *p* = 0.632 for EPA at 10, 30, 60 and 100 µM, respectively) (Figure 4A). In fact, the effect of EPA was found from 10 µM but did not increase anymore for higher concentrations. By contrast, supplementation with EPA markedly reduced the expression of *MMP-13* induced by PA and a high glucose level from the lowest concentration used (10 µM) (Figure 4B). Moreover, EPA restored the expression of *type 2 collagen*, but without statistical significance (Figure 4C), and it did not change the inhibitory effect of PA on the mRNA level of *aggrecan* whatever the dose used (Figure 4D).

Interestingly, EPA exhibited strong anti-inflammatory activities by lowering *COX-2* (Figure 5A) and *mPGES* (Figure 5B) mRNA levels, the expression of *mPGES* reaching those found in untreated control chondrocytes. As expected, these changes led to a four-fold reduction in the rate of PGE_2_ when cells challenged with PA and a high glucose level were supplemented with 30 µM EPA (Figure 5C).

### 2.4. Identification of Receptors Involved in PA and EPA Effects

In order to identify FFARs activated by PA and EPA in hyperglycemic conditions, we examined the effects of agonists (GW 9508 and TUG 891) and antagonists (DC 260126 and AH 7614) on the COX-2/PGE_2_ pathway found to be the most significantly modulated by PA and EPA.

When used alone, FFAR ligands slightly affected the *COX-2* mRNA level. Both FFARs antagonists (DC 260126 and AH 7614) up-regulated the gene encoding COX-2 (Figure 6B and Figure 7B), while the FFAR4 agonist (TUG 891) reduced its expression (Figure 7A), the agonist of FFRA1 (GW 9508) being ineffective (Figure 6A). The data also indicated that EPA remained unable to stimulate the expression of *COX-2* even upon supplementation with FFAR ligands.

By contrast, FFAR ligands affected the chondrocyte response to PA. Those specific for FFAR1 reduced the overexpression of *COX-2* in PA-treated cells, GW 9508 being the most effective (Figure 6A,B), while both FFAR4 ligands exhibited the opposite effect. TUG 891 increased the PA-induced *COX-2* up-regulation when compared to the corresponding control (Figure 7A), but AH 7614 reduced it (Figure 7B). Moreover, most of the FFAR ligands did not change the beneficial effect of EPA on the chondrocyte response to PA. In both cases (with or without ligands), EPA reduced the PA-stimulated expression of *COX-2* to the corresponding control level (Figure 6A,B and Figure 7A,B). Only AH 7614 decreased the protective activity of EPA as the *COX-2* mRNA level did not reach the unstimulated control level (Figure 7B).

The data from the PGE_2_ assays in culture supernatants were generally consistent with those from qPCR analysis, except for some conditions (Figure 6C,D and Figure 7C,D). The inhibitory effect of EPA on PA-induced *COX-2* expression still found upon addition of GW 9508 was not shown for PGE_2_ release (Figure 6A,C). Similarly, a marked increase in PGE_2_ synthesis was noticed when chondrocytes were challenged with PA and/or EPA in the presence of AH 7614, while the *COX-2* mRNA level was barely increased over control value (Figure 7B,D). 

## 3. Discussion

The present study aimed to gain insight into the role of two components of the MetS, i.e., dyslipidemia and hyperglycemia, in the pathogenesis of OA. Our data indicated that lipids modulated chondrocyte functions depending on the nature of the free fatty acids. We showed that PA displayed deleterious effects in cultured rat articular chondrocytes, which were further increased by hyperglycemia. Interestingly, EPA had a beneficial impact by suppressing the inflammatory chondrocyte response to PA and glucose, and by reducing *MMP-13* expression to the untreated control’s level. FFAR4 partly mediated the anti-inflammatory effects of EPA and the activation of FFAR1 strongly reduced the inflammatory effects of PA associated with high glucose.

Recent findings indicated that dyslipidemia is likely one of the most important factors linking MetS and OA. A meta-analysis reported a higher risk of dyslipidemia in patients with knee or hand OA [28]. More specifically, dietary saturated fatty acids intake was associated with increased radiographic progression of knee OA [23] with a strong link between the fatty acid chain length and the end-stage OA [29]. Experimental studies provide evidence for a key role of saturated fatty acids in OA, as they were shown to promote deleterious effects for joint structures. For instance, Sekar et al. reported that rats fed with saturated fatty acids developed, in parallel, signs of MetS and OA-like knee changes [22]. A high fat diet rich in saturated fatty acids also increased the severity of post-traumatic OA [21,30,31]. Among saturated fatty acids found in the synovial fluid from OA patients, PA was the most abundant [15]. Higher serum levels of this fatty acid were associated with more severe radiographic knee OA [17], and equine OA joints can be distinguished from normal joints based on elevated articular levels of PA in extracellular vesicles-enriched pellets [32]. In addition, PA was shown to be the most effective in rats to induce degenerative changes in articular cartilage [22], and a high-fat diet with a high amount of PA reduced chondrocyte response to Insulin Growth Factor-1 (IGF-1) [33]. As found in the present study, most of the in vitro experiments using cultured chondrocytes or cartilage explants from animal, indicated that treatment with PA resulted in the expression and/or the secretion of pro-inflammatory factors and in cartilage matrix degradation. However, the results of studies examining the effects of PA in human articular chondrocytes are still equivocal. For instance, Sekar et al. reported an increase in the gene expression of degradative proteins, i.e., MMP-13 and A Disintegrin and Metalloproteinase with ThromboSpondin motifs-4 (ADAMTS-4) and a decrease in the mRNA levels of aggrecan [22], while Alvarez-Garcia et al. failed to detect any change in the expression of aggrecan and MMP-13 upon treatment of human chondrocytes with PA [34]. In another study, PA was unable to affect the expression of *COX-2* nor PGE_2_ production and displayed a rather protective effect against the glycosaminoglycans release from human cartilage explants [35]. Interestingly, Frommer et al. observed that the response of synovial fibroblast to fatty acids was rather dependent on the patient than on the joint disease [36]. As stressed herein, some disorders associated with MetS such as hyperglycemia may alter chondrocyte response to PA, and so may explain the discrepancies between various studies in humans.

In addition to dyslipidemia, type 2 diabetes is strongly predictive of OA severity, suggesting the involvement of hyperglycemia. As many tissues and organs, articular cartilage requires glucose to maintain extracellular matrix homeostasis. This sugar plays a role in signaling pathways and regulates metabolic functions both in health and diseases. By contrast to normal human chondrocytes which are able to modulate GLUT 1 synthesis and degradation in response to excessive glucose, OA-derived chondrocytes accumulated more glucose and produced more reactive oxygen species known to mediate many of the effects induced by inflammatory cytokines [37]. Moreover, the exposure of normal human chondrocytes to high glucose up-regulated the expression of *MMP-1* [38]. Another study also demonstrated that hyperglycemia induced the protein expression of COX-2 and the production of PGE_2_, IL-6 and MMP-13, and concomitantly decreased the protein expression of type 2 collagen in human chondrocytes [39]. As shown previously with mouse chondrocytes [40] and even if we found a slight increase in the mRNA level of *mPGES* under hyperglycemic conditions, we failed to detect any significant change in the expression of genes encoding inflammatory or degenerative proteins in healthy rat chondrocytes cultured with a high glucose level, suggesting that the response of healthy chondrocyte to high glucose alone may therefore be species-dependent. The data from various studies indicated that elevated concentrations of glucose may also influence the effects of growth factors or pro-inflammatory cytokines. Exposure of human chondrocytes to high glucose prevented Transforming Growth Factor (TGF)-induced down-regulation of *MMP-13* gene expression [38] and the response of mouse chondrocytes to IL-1β increased when cultured in a high glucose environment [40]. Kelley et al. also observed that hyperglycemia impaired the ability of the IGF-1 to stimulate proteoglycan synthesis in cultured rabbit chondrocytes [41]. In agreement with these findings, we reported that glucose at a high level further enhanced the inflammatory and degenerative effects induced by PA in rat chondrocytes, suggesting that both dyslipidemia and hyperglycemia may contribute in combination to cartilage damage during OA.

Free fatty acids display various biological activities depending on their length and degree of saturation. By contrast to saturated fatty acids, n-3 PUFAs have often been described as anti-inflammatory, and so play an important role in chondrocyte and cartilage homeostasis. Surgically induced OA in Fat-1 mice, which are able to endogenously convert n-6 PUFAs to n-3 PUFAs, was shown to be less severe when compared to wild-type operated mice, with reduced expression of MMP-13 and ADAMTS-5 [25,42]. More especially, intra-articular EPA-incorporating gelatin hydrogels prevented OA progression through the inhibition of both IL-1β and MMP-13 expression [43]. Many studies have investigated the effects of n-3 PUFAs on chondrocyte responses to various pro-inflammatory cytokines. The IL-1α-induced expression of COX-2, MMP-3 and -13, ADAMTS-4 and -5, IL-1α, IL-1β and TNFα was reduced upon pre-incubation of bovine chondrocytes with EPA [44], and EPA decreased the IL-1β-mediated release of glycosaminoglycans from bovine cartilage explants [45]. However, conflicting results have been reported mainly due to species differences and experimental conditions differing in the time of exposure to fatty acids, the nature of the inflammatory stimulus and the fatty acids concentrations. For instance, EPA was not able to change the IL-1β-induced expression of *COX-2* and *MMPs* in canine chondrocytes, but decreased the release of NO, which is a key mediator in OA pathogenesis [46]. At present, no study aimed to evaluate the effects of a saturated fatty acid together with a n-3 PUFA in chondrocytes. Here, we showed for the first time that EPA was able to reverse the pro-inflammatory effects of PA and hyperglycemia in those cells. Various mechanisms may contribute to the anti-inflammatory activity of EPA. The elevated production of PGE_2_ in chondrocytes cultured with PA and high glucose resulted from the activation of the arachidonic acid cascade, notably through the action of COX-2 and mPGES. As EPA and arachidonic acid are homologues and compete for enzymes that produce eicosanoids, it is likely that the beneficial effect of EPA may be related to its ability to inhibit the production of n-6 PUFA-derived prostaglandins and leukotrienes, which are described as pro-inflammatory [47]. In addition, EPA may serve as precursor for anti-inflammatory specialized pro-resolving lipid mediators (SPMs) such as resolvins and protectins. SPMs function as signaling molecules that promote the restoration of vascular integrity, tissue regeneration and repair, and decrease or resolve inflammation by inhibiting inflammatory lipid mediators and cytokines [48].

Long-chain saturated and unsaturated fatty acids including PA and EPA, respectively, also trigger by themselves signaling pathways through binding to and the activation of FFAR1 and 4. Based on FFARs-deficient mouse models, FFAR1 and 4 were also identified as key signaling receptors involved in articular cartilage homeostasis [49,50]. We showed herein that the binding of TUG 891 to FFAR4 and both FFAR1 and 4 antagonists modulated the mRNA level of *COX-2* in unstimulated rat chondrocyte cultured under hyperglycemic conditions, suggesting that both FFAR1 and 4 may be involved in the chondrocyte inflammatory response. TUG 891 was previously shown to inhibit the expression of pro-inflammatory cytokines and to rescue the expression of *type 2 collagen* and *aggrecan* in IL-1β- stimulated ATDC5 chondrocyte [51]. Notwithstanding the above, the data from our experiments with PA, EPA and FFAR ligands are rather difficult to analyze. Firstly, while agonists and antagonists are expected to have opposite activity, we showed unexpectedly that both FFAR1 ligands reduced the inflammatory effect of PA in hyperglycemic conditions. Secondly, the FFAR4 agonist TUG 891 with anti-inflammatory activity increased the inflammatory chondrocyte response to PA. By contrast, the pro-inflammatory FFAR4 antagonist AH 7614 reduced the stimulatory effect of PA and high glucose on *COX-2* expression. In fact, the identification of the receptor involved in cell response to fatty acids is rather complex since the activation of FFARs is known to be coupled to multiple downstream effectors [52]. As a result and according to the concept of “biased agonism”, different ligands may impart distinct signaling and biological attributes to a given receptor [53]. In any case, our data further support the binding of PA to FFAR1 as the FFAR1 ligand GW 9508 inhibited the PA-induced *COX-2* expression in rat chondrocyte cultured under hyperglycemic conditions (Figure 8). Such a beneficial role of GW 9508 was also reported in a human chondrocyte cell line stimulated with advanced-glycation end-products. This FFAR1 agonist reduced the release of pro-inflammatory cytokines and protected type 2 collagen and aggrecan from degradation by down-regulating the expression of degenerative proteins [54]. Our findings also provide evidence for a key role of FFAR4 but not FFAR1 in mediating the anti-inflammatory effect of EPA in rat chondrocytes since AH 7614 alleviated the effects of EPA on the chondrocyte response to PA in high glucose conditions.

Our study is the first to assess the effects on cartilage cells of an unsaturated free fatty acid (EPA) on those induced by a saturated fatty acid (PA). In order to provide sound conclusions, we chose to use healthy articular chondrocytes to prevent pre-existing metabolic disorders that may result in impaired cell functions and thus may alter chondrocyte response to fatty acids and glucose. Interestingly, our findings indicate that young and healthy articular cells exhibit an inflammatory and degenerative response to PA, which may be worsened by hyperglycemia, suggesting that chondrocyte functions may be altered early during MetS development. Additional in vitro experiments using human chondrocytes from OA patients would be useful to better understand the contribution of FFAs in MetS-associated OA. More specifically, it would be interesting to evaluate the chondrocyte response to pro-inflammatory factors according to the metabolic status of the patients. It would also be interesting to evaluate the effect of EPA on the spontaneous expression of inflammatory and degenerative proteins in chondrocytes originated from OA patients with various metabolic disorders. It is noteworthy that the use of PA in isolation may be a limitation of the study as the in vitro effects induced by a single fatty acid may not be biologically relevant. In vivo, the tissue fatty acid profiles are composed of a complex mixture and some adverse effects may be reversed by others. In addition, the use of a single fatty acid does not reflect the actual biological state within the joint and makes the extrapolation of the effects to the whole joint difficult. Although we found that PA and EPA alter chondrocyte functions, it is likely that these fatty acids and their interactions may have different effects on cartilage when applied to the whole joint. Further experiments with fatty acids mixture would therefore be helpful to understand the role of dyslipidemia in OA. A more comprehensive clinical evaluation of the articular lipid content would also be useful to better mimic the cross-talk between MetS and OA.

In conclusion, the present study offers new perspectives to better understand the contribution of dyslipidemia and hyperglycemia to the chronic low-grade inflammation found in OA and explains why dietary imbalance with an excess of saturated fatty acids and a low level in n-3 PUFAs may disrupt cartilage homeostasis. By elucidating the role of metabolic stress in OA, we also gain insights into novel molecular targets for disease-modifying treatments. Our findings suggest, for instance, that nutritional interventions aimed at increasing the relative abundance of PUFAs or their derivatives could safely reduce the inflammation underlying cartilage degeneration in OA.

## 4. Materials and Methods

### 4.1. Isolation and Culture of Rat Chondrocytes

Normal articular cartilage was obtained from 6-week-old male Wistar rats (130 to 150 g) (Charles River, Lyon, France) in accordance with the ARRIVE guidelines and the Directive 2010/63/EU. Articular cartilage was aseptically dissected from femoral head caps and then cut into small pieces. Chondrocytes were isolated by sequential digestion of the extracellular matrix with pronase (0.15%, *w*/*v*) for 2 h and collagenase B (0.2%, *w*/*v*) (Sigma-Aldrich, Saint-Quentin-Fallavier, France) overnight at 37 °C. After centrifugation of the resulting cells and suspension in Dulbecco’s Modified Eagle’s medium/Ham’s F-12 (DMEM/Ham’s F-12) supplemented with 2 mM L-glutamine, 50 µg/mL penicillin-streptomycin and 10% (*v*/*v*) heat-inactivated fetal calf serum (FCS), cells were seeded at high density (10^4^ cells/cm^2^). Cells were cultured at 37 °C until confluence in a humidified atmosphere containing 5% CO_2_. All experiments were performed with first-passage chondrocytes.

### 4.2. Treatment of Rat Chondrocytes with Fatty Acids, Glucose and FFAR Ligands

Analytical-grade sodium palmitate, EPA and bovine serum albumin (BSA, fatty acid free) were purchased from Sigma-Aldrich (France). GW9508, TUG891, DC260126 and AH7614 were purchased from Cayman (Ann Arbor, MI, USA).

PA was prepared as a fatty acid–BSA complex. The fatty acid was dissolved in DMEM at 70 °C to yield a 2 mM stock solution. This stock solution was then diluted 1:1 in 340 µM BSA made with DMEM to yield a final 1 mM PA solution. All solutions were warmed at 37 °C before they were mixed together. The mix was further incubated at 37 °C until the solution became clear. EPA was prepared into a 100 mM solution in ethanol and FFARs agonists (GW9508; TUG891) and antagonists (DC260126; AH7614) were dissolved in DMSO to yield a final 1 mM stock solution. All stock solutions were diluted the day of use with the appropriate medium.

The articular chondrocytes were maintained in DMEM low glucose (1 g/L) supplemented with 2 mM L-glutamine, 50 µg/mL penicillin–streptomycin and 1% FCS overnight prior to incubation with conditional media. Chondrocytes were then incubated with or without the PA–BSA complex used at different concentrations ranging from 0.1 to 0.5 mM. DMEM with glucose at 4.5 g/L was used to perform experiments in high glucose conditions. For some experiments, the high glucose-containing medium was supplemented with EPA used at different concentrations ranging from 10 to 100 µM. In the experiments aimed to investigate the contribution of FFARs, agonists and antagonists (30 µM) were added 4 h before treatment with PA and/or EPA. Vehicle controls were prepared in the same way as the PA-BSA complex except ethanol and/or DMSO was used instead of PA solution. At the end of the treatment (24 or 48 h), the cell lysates were harvested for gene expression analysis and the culture supernatants were centrifuged 5 min at 600 g and kept at −80 °C until PGE_2_ assays.

### 4.3. RNA Isolation and Real-Time Polymerase Chain Reaction

Total RNA was extracted from cultured chondrocytes using RNeasyplus Mini Kit^®^ (Qiagen, Courtaboeuf, France) according to the manufacturer’s instructions. RNA concentration was determined by measurements of the optical density at 260 nm on a NanoDrop ND-1000 Spectrophotometer (Labtech, Dutsher, Bernolsheim, France). RNA samples were then reverse-transcribed for 90 min at 37 °C using oligo-dT primers (100 pmol) and Moloney Murine Leukemia Virus reverse transcriptase (M-MLV) reverse transcriptase (200 U) (Invitrogen, ThermoScientific, Waltham, MA, USA). cDNAs production was performed in a Mastercycler gradient thermocycler (Eppendorf, Montesson, France). Gene expression was analyzed by quantitative real time PCR (Via 7 ^TM^, Applied Biosystems, ThermoScientific, Waltham, MA, USA) using the SYBRgreen^TM^ master mix system (Biorad, Marnes-la-Coquette, France) according to the protocol provided. The gene-specific primer pairs optimized for this method are listed in Table 1. As a control of the amplification specificity, melting curve analysis was performed for each PCR experiment to separate the specific product from the non-specific products (if any). Amplified products were also visualized by electrophoresis on a 1% agarose gel stained with Gel Red (Biotium, Interchim, Montluçon, France). Each run included positive and negative reaction controls. The expression level of target genes was normalized to that of the housekeeping gene coding the ribosomal protein 29 (RPS29) measured in parallel samples. Quantification was achieved using the 2^−ΔΔCt^ method and the results were expressed as fold expression over the appropriate control.

### 4.4. PGE_2_ Determination by Enzyme-Linked Immunosorbent Assay (ELISA)

Secreted PGE_2_ concentrations in conditioned media from the cultured chondrocytes were determined by a sequential competitive ELISA using microplate wells coated with a goat anti-mouse polyclonal antibody and a mouse monoclonal antibody to PGE_2_ (Prostaglandin E_2_ Parameter Assay Kit, Bio-Techne, Noyal-Chatillon/Seiche, France). In a first step, PGE_2_ in the samples was allowed to bind to the primary antibody. In the second step, horseradish peroxidase (HRP)-labeled PGE_2_ was bound to the remaining antibody sites. The absorbance at 450 nm was read on a Varioskan Flash microplate reader (ThermoScientific, Waltham, MA, USA) and a standardized 4-PL curve was used to obtain the absolute values. The detection limit was 31 pg/mL and the intraassay and interassay variations were 6.7% and 10.6%, respectively.

### 4.5. Statistical Analysis

Cell treatments were performed in triplicate and repeated three times independently. Results were expressed as the mean ± SEM of triplicate independent experiments. Statistical analyses were performed using GraphPad Prism v5 software using one-way ANOVA followed by either the Tukey’s multiple comparisons test or the unpaired t test, with Welsh’ correction when variances were significantly different. *p* values less than 0.05 were considered significant.

## Figures and Tables

**Figure 1 ijms-25-01810-f001:**
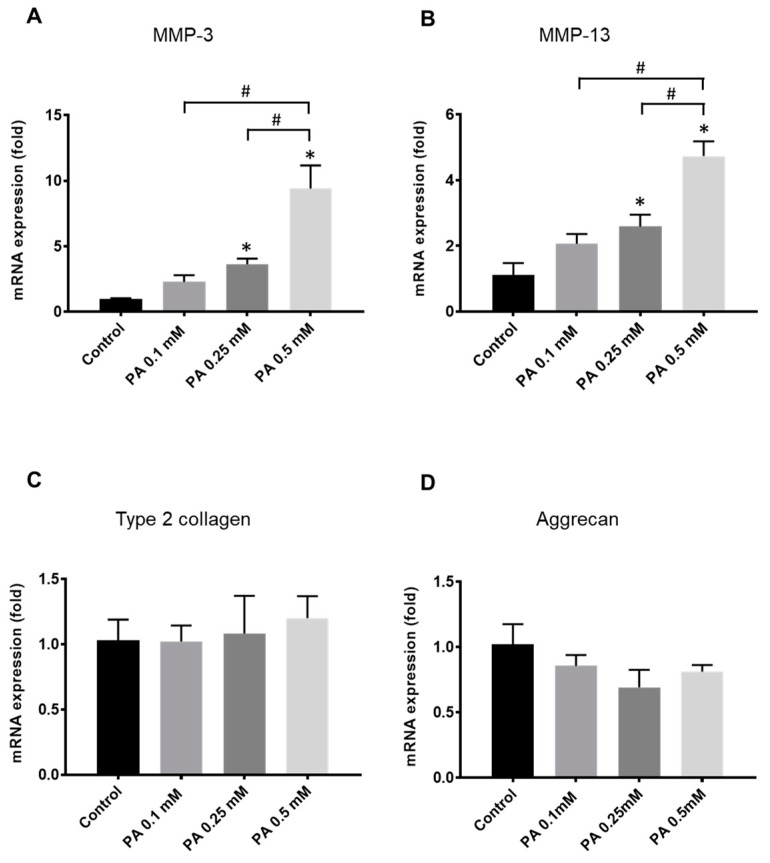
Palmitic acid (PA) dose-dependently up-regulates the gene expression of degradative enzymes (metalloproteinases-3 and -13, *MMP-3* and -*13*) and modulates the expression of *aggrecan* but not that of *type 2 collagen*. Articular rat chondrocytes were cultured in normal glucose conditions (1 g/L) with vehicle (control) or increasing concentrations of PA ranging from 0.1 to 0.5 mM. mRNA levels for *MMP-3* (**A**), *MMP-13* (**B**), *type 2 collagen* (**C**) and *aggrecan* (**D**) were determined by quantitative PCR in cell lysates collected 24 h after treatment with PA. The data represent mean ± SEM of three independent experiments and are expressed as fold change compared to the untreated control group. * *p* < 0.05 vs. untreated control group, # *p* < 0.05 vs. PA 0.5 mM.

**Figure 2 ijms-25-01810-f002:**
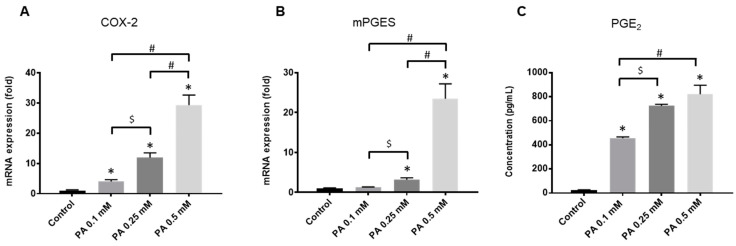
Palmitic acid (PA) induces an inflammatory response in rat chondrocytes. Articular cells were cultured in normal glucose conditions (1 g/L) with vehicle (control) or increasing concentrations of PA ranging from 0.1 to 0.5 mM. Cell lysates were harvested at 24 h and culture supernatants were collected at 48 h. mRNA levels for *cyclooxygenase-2 (COX-2)* (**A**) and *microsomal prostaglandin-E synthase (mPGES)* (**B**) were determined by quantitative PCR, and Prostaglandin-E_2_ (PGE_2_) released in culture media was measured by ELISA (**C**). The data represent mean ± SEM of three independent experiments and are expressed as fold change compared to the untreated control group for mRNA expression. * *p* < 0.05 vs. untreated control group, $ *p* < 0.05 vs. PA 0.25 mM, # *p* < 0.05 vs. PA 0.5 mM.

**Figure 3 ijms-25-01810-f003:**
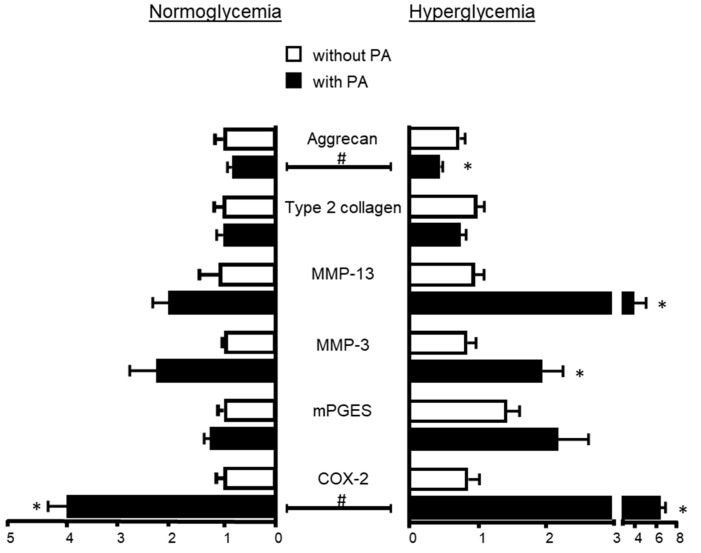
Glucose at high levels increases the chondrocyte response to palmitic acid (PA). Rat chondrocytes were cultured for 24 h with PA (0.1 mM) in a medium containing either normal (1 g/L) or high (4.5 g/L) glucose level. The expression of the gene encoding metalloproteinases-3 and -13 (MMP-3 and -13), type 2 collagen, aggrecan, cyclooxygenase-2 (COX-2) and microsomal prostaglandin-E synthase (mPGES) was determined by quantitative PCR. The data represent mean ± SEM of three independent experiments and are expressed as fold change compared to the control group without PA. * *p* < 0.05 vs. untreated control group in the same glucose conditions, # *p* < 0.05 vs. normoglycemia conditions.

**Figure 4 ijms-25-01810-f004:**
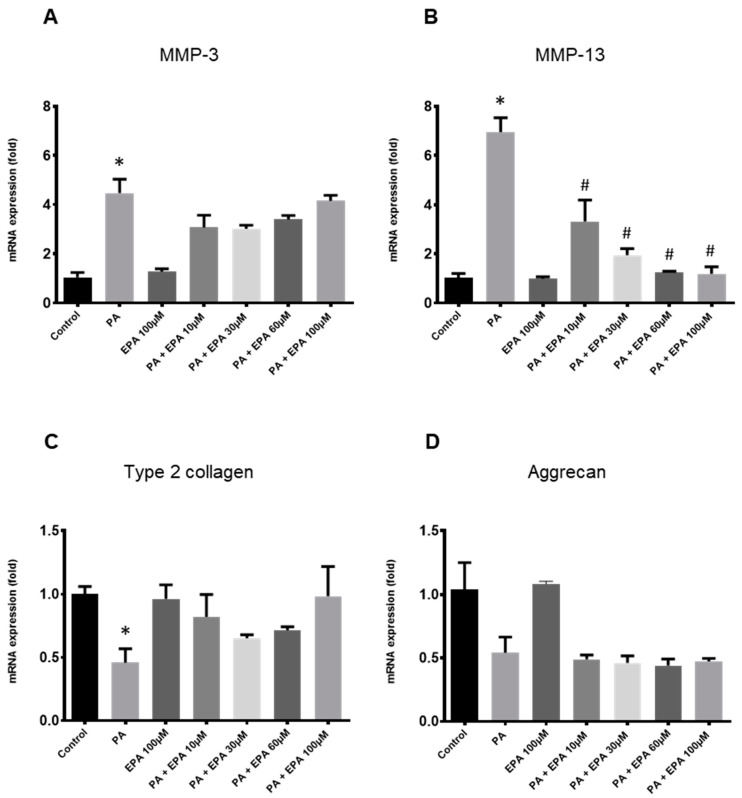
Eicosapentaenoic acid (EPA) protects rat chondrocytes from the deleterious effects of palmitic acid (PA) and hyperglycemia. Articular rat chondrocytes were cultured for 24 h in a medium containing high glucose level (4.5 g/L) with vehicle (control) or PA (0.1 mM) with or without various concentrations of EPA ranging from 10 to 100 µM. mRNA levels for *MMP-3* (**A**) and -*13* (**B**), *type 2 collagen* (**C**) and *aggrecan* (**D**) were determined by quantitative PCR. The data represent mean ± SEM of three independent experiments and are expressed as fold change compared to the untreated control group. * *p* < 0.05 vs. untreated control group, and # *p* < 0.05 vs. PA-treated group.

**Figure 5 ijms-25-01810-f005:**
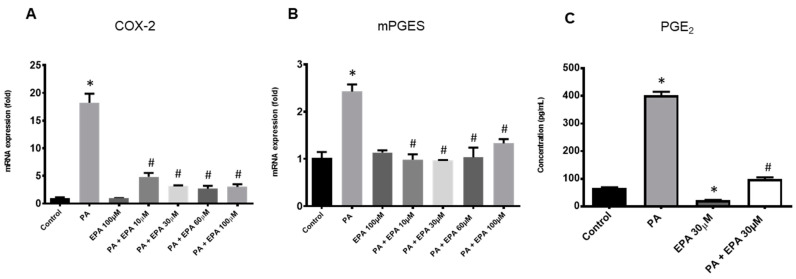
Eicosapentaenoic acid (EPA) strongly reduces the inflammatory chondrocyte response to palmitic acid (PA) and hyperglycemia. Articular rat chondrocytes were cultured in a medium containing high glucose level (4.5 g/L) with vehicle (control) or PA (0.1 mM) with or without various concentrations of EPA ranging from 10 to 100 µM. Cell lysates were harvested at 24 h and culture supernatants were collected at 48 h. mRNA levels for *cyclooxygenase-2 (COX-2)* (**A**) and *microsomal prostaglandin-E synthase (mPGES)* (**B**) were determined by quantitative PCR, and Prostaglandin-E_2_ (PGE_2_) released in culture media was measured by ELISA (**C**). The data represent mean ± SEM of three independent experiments and are expressed as fold change compared to the untreated control group. * *p* < 0.05 vs. untreated control group, and # *p* < 0.05 vs. PA-treated group.

**Figure 6 ijms-25-01810-f006:**
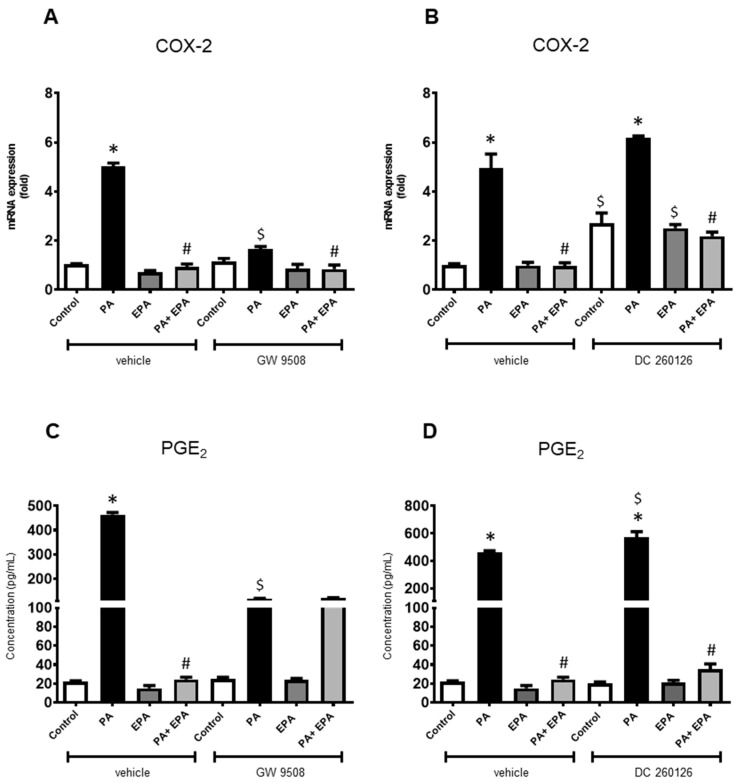
The FFAR1 agonist (GW 9508) and antagonist (DC260126) reduce the inflammatory chondrocyte response to palmitic acid (PA) and hyperglycemia. Rat chondrocytes were cultured in a medium containing high glucose level (4.5 g/L). The articular cells were incubated with vehicle (control) or FFAR1 ligands (30 µM) 4 h before treatment with or without PA (0.1 mM) and/or eicosapentaenoic acid (EPA) (30 µM). Cell lysates were harvested at 24 h and culture supernatants were collected at 48 h. mRNA levels for *cyclooxygenase-2 (COX-2)* (**A**,**B**) were determined by quantitative PCR, and Prostaglandin-E_2_ (PGE_2_) released in culture media was measured by ELISA (**C**,**D**). The data represent mean ± SEM of three independent experiments and are expressed as fold change compared to the untreated vehicle control group. * *p* < 0.05 vs. untreated control group, # *p* < 0.05 vs. PA-treated group, and $ *p* < 0.05 vs. vehicle group.

**Figure 7 ijms-25-01810-f007:**
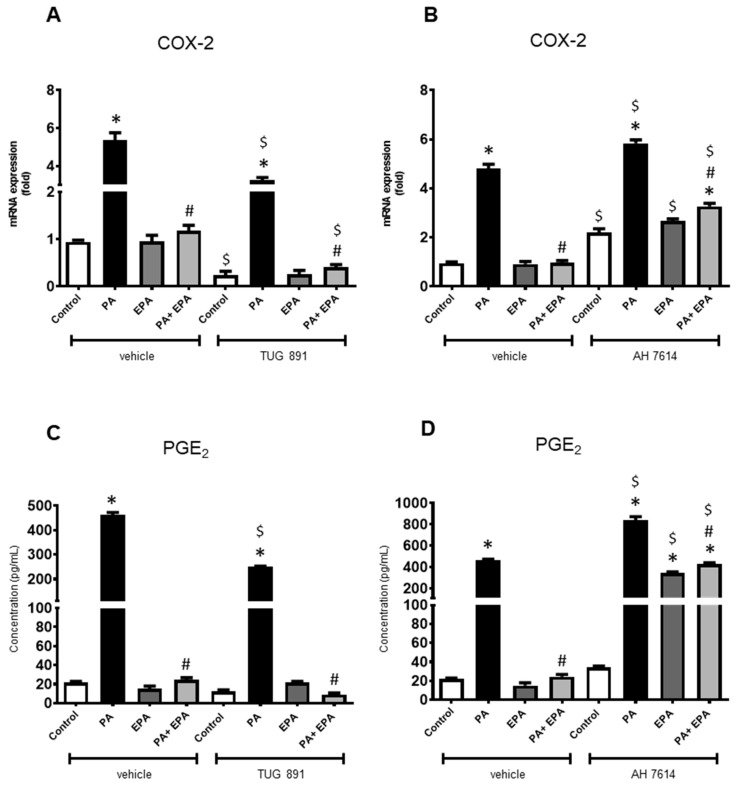
FFAR4 antagonism alleviates the protective effect of eicosapentaenoic acid (EPA) on the inflammatory chondrocyte response induced by palmitic acid (PA) and hyperglycemia. Rat chondrocytes were cultured in a medium containing high glucose level (4.5 g/L). The articular cells were incubated with vehicle (control) or FFAR4 ligands (TUG 891 as agonist and AH 7614 as antagonist) (30 µM) 4 h before treatment with or without PA (0.1 mM) and/or EPA (30 µM). Cell lysates were harvested at 24 h and culture supernatants were collected at 48 h. mRNA levels for *cyclooxygenase-2 (COX-2)* (**A**,**B**) were determined by quantitative PCR, and Prostaglandin-E_2_ (PGE_2_) released in culture media was measured by ELISA (**C**,**D**). The data represent mean ± SEM of three independent experiments and are expressed as fold change compared to the untreated vehicle control group. * *p* < 0.05 vs. untreated control group, # *p* < 0.05 vs. PA-treated group, and $ *p* < 0.05 vs. vehicle group.

**Figure 8 ijms-25-01810-f008:**
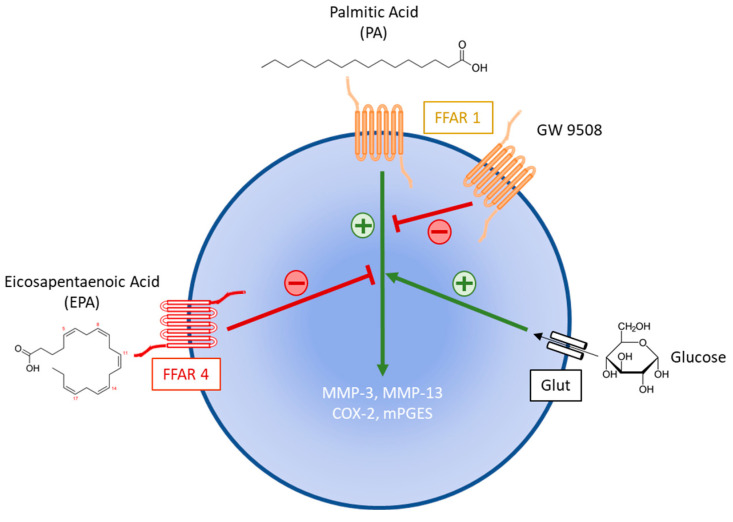
Potential mechanisms involved in the regulation by glucose and FFAR ligands (EPA and GW 9508) of the stimulatory effect of palmitic acid on the expression of degenerative (MMP-3 and -13) and inflammatory (COX-2 and mPGES) genes. Green arrows denote activation whereas red lines indicate inhibition.

**Table 1 ijms-25-01810-t001:** Sequences of specific primers for RT-PCR analyses.

Genes	Sequences 5′-3′
*Aggrecan*	Fwd: CAA-CCT-CCT-GGG-TGT-AAG-GA
	Rev: TGT-AGC-AGA-TGG-CGT-CGT-AG
*Collagen 2*	Fwd: TCC-CTC-TGG-TTC-TGA-TGG-TC
	Rev: CTC-TGT-CTC-CAG-ATG-CAC-CA
*COX-2*	Fwd: TAC-AAG-CAG-TGG-CAA-AGG-CC
	Rev: CAG-TAT-TGA-GGA-GAA-CAG-ATG-GG
*mPGES*	Fwd: ACC-CTC-TCA-TCG-CCT-GGA-TA
	Rev: ATG-CGT-GGG-TTC-ATT-TTG-CC
*MMP-3*	Fwd: TCT-GGG-CTA-TCC-GAG-GTC-AT
	Rev: TGC-ATC-GAT-CTT-CTG-GAC-GG
*MMP-13*	Fwd: TCT-GGG-CTA-TCC-GAG-GTC-AT
	Rev: TGC-ATC-GAT-CTT-CTG-GAC-GG
*RP29*	Fwd: CTC-TAA-CCG-CCA-CGG-TCT-GA
	Rev: ACT-AGC-ATG-ATT-GGT-ATC-AC

Collagen 2—type 2 collagen; COX-2—cyclooxygenase type 2; mPGES—microsomal prostaglandin E_2_ synthase; MMP-3 & -13—matrix metalloproteinase 3 & 13; RP29—ribosomal protein 29; Fwd—forward primer; Rev—reverse primer; RT-PCR—real-time polymerase chain reaction.

## Data Availability

Data is contained within the article.

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
