# Peer review of "Beneficial Impact of Eicosapentaenoic Acid on the Adverse Effects Induced by Palmitate and Hyperglycemia on Healthy Rat Chondrocyte"

_ijms, 2024, doi:10.3390/ijms25031810_

Round 1
Reviewer 1 Report
Comments and Suggestions for Authors
Fig 4 – in section C the sample PA results significant whereas in D is not. Why ?
Fig.5 C – still statistics. I don’t understand the same significance (*) in PA and EPA30 versus controls. It looks like quite different.
In general, p<0.05 in all figures seems poor significant respect to real differences of graph. Are you sure about the statistic calculation ?
Author Response
Reviewer #1:
Fig 4 – in section C the sample PA results significant whereas in D is not. Why ?
We agree that it seems to be within very close range. However, the standard deviation for the control group in figure 4D is greater than that found for figure 4C. Consequently, the p-value for the comparison between the control and PA groups is statistically significant for figure 4C (p=0.012) but not for figure 4D (p=0.112).
Fig.5 C – still statistics. I don’t understand the same significance (*) in PA and EPA30 versus controls. It looks like quite different.
In general, p<0.05 in all figures seems poor significant respect to real differences of graph. Are you sure about the statistic calculation ?
We did double check our data and can claim we are indeed certain about our statistic calculation. In fact, in order to improve the readability of the figures, we chose not to put several symbols (*, **, *** for instance) to indicate the different levels of significance. This would explain why we marked in Fig5C a same * even though p<0.0001 for PA and p=0.007 for EPA30 compared to control.
For the sake of clarity, and because it has not been mentioned by other reviewers, we choose to keep our initial representation. However, if the reviewer stands firm that should be done, we will be happy to provide a final version with distinct statistical analysis in adjusted figures.

Reviewer 2 Report
Comments and Suggestions for Authors
Your work need more experiments to enhance the novelty and support your goals and aims
1- Please, detect the effect of EPA on Insulin, HOMA-IR, and Adiponectin
2- Please, detect the effect of EPA on weight.
3- Please, detect some novel targeted genes and noncoding RNAs, LncRNA and microRNAs.
4- Your invivo study not confirmed by histopathology and immunohistochemistry to support your conclusion.
5- Crossover your study you just uesd RT-PCR and ELISA techniques, as mentioned before you can use at least histopathology and IHC. If possible western blotting to detect the protein expressions of your targeted genes.
Author Response
Reviewer #2:
Your work need more experiments to enhance the novelty and support your goals and aims
1- Please, detect the effect of EPA on Insulin, HOMA-IR, and Adiponectin
2- Please, detect the effect of EPA on weight.
3- Please, detect some novel targeted genes and noncoding RNAs, LncRNA and microRNAs.
4- Your invivo study not confirmed by histopathology and immunohistochemistry to support your conclusion.
5- Crossover your study you just uesd RT-PCR and ELISA techniques, as mentioned before you can use at least histopathology and IHC. If possible western blotting to detect the protein expressions of your targeted genes.
We thank the reviewer for his/her questions and understand his/her concern regarding our work.
If performing the analysis suggested by the reviewers in questions 1, 2, 4 and 5 would definitely complement our current data, there would necessitate to do the experiments in an animal model. Our study has been achieved in cell culture since our goal was to tackle our hypothesis through in vitro work only. Indeed, it is somehow difficult to identify specific and/or combined role of hyperglycemia and dyslipidemia in an exclusive in vivo model. And we sought to answer our initial hypothesis in vitro before possibly going further with in vivo model(s). If we do agree that could be considered as a limitation of our current work (see before-the-last paragraph of the discussion) and that in vivo experiments would be further required to substantiate our current work, the latter are beyond the scope of our study.
Regarding the suggestion for the detection of non-coding RNAs, LncRNA and microRNAs (question 4), this is undoubtedly an interesting and innovative idea. However, unless we focus, with potential bias, on speculative candidates that have been identified in other types of cells under high glucose/hyperglycemic and/or PA/dyslipidemic conditions, this would need first to generate our own specific list of candidates through high-throughput technology (transcriptomic and/or proteomic analysis). We are afraid that it is not be possible within the frame of this work and before the resubmission deadline.

Reviewer 3 Report
Comments and Suggestions for Authors
The study provides a comprehensive investigation into the impact of metabolic stresses, specifically dyslipidemia and hyperglycemia, on OA pathogenesis. The focus on chondrocyte response and the potential protective role of EPA adds valuable insights to the understanding of OA at the molecular level.
The experimental design, involving the culture of rat chondrocytes with PA and/or EPA in normal or high glucose conditions, is well-structured. The use of qPCR to analyze gene expression and enzyme-linked immunosorbent assay to measure PGE2 release ensures a detailed examination of the molecular changes occurring in response to metabolic stresses. This reviewer finds it valuable to include other measures such as protein content (via Western blot or ELISA) and/or flow cytommetry to help support the findings throughout. The qPCR readouts makes this reviewer wonder whether there were any changes to cell growth and would suggest the inclusion of some photographic images depicting each treatment condition to 'break up' the mRNA expression data presented & provide insight into whether cell growth/death could potentially impact the cell yield for qPCR. Another suggestion following on from these experiments would be to use an alternative readout via ddPCR to provide an absolute measure of copies/droplet (ng or RNA).
The findings regarding the up-regulation of MMP-3 and -13 by PA, along with the induction of COX-2 and mPGES, shed light on the inflammatory and degenerative processes associated with dyslipidemia and hyperglycemia. The exacerbating effect of high glucose concentration on chondrocyte response to PA underscores the potential synergy between these metabolic stresses in contributing to OA. It would be interesting to replicate these experiments in human cultured chondrocytes (a more relevant clinical model) to see if the data can be replicated or subject to interspecies variability.
The protective effects of EPA are a significant highlight of the study. The suppression of inflammatory effects induced by PA and glucose, coupled with a strong reduction in MMP-13 expression, suggests a potential therapeutic role for n-3 polyunsaturated fatty acids in preserving chondrocyte functions and maintaining cartilage homeostasis.
The identification of FFAR4 and FFAR1, in mediating EPA effects adds depth to the mechanistic understanding of the observed responses. The clarification that dyslipidemia associated with hyperglycemia may contribute to OA pathogenesis provides a valuable link between metabolic syndrome components and joint health. A minor suggestion would be to include a mechanism diagram for figure 6 to summarize the antagonistic roles & influences on biomarkers measured throughout this study.

Comments on the Quality of English LanguageThe manuscript is well written with minor edits in the attached PDF file. I would suggest improving the references included in the background to better introduce the current study. This reviewer would also like a future implications and impactful segment towards the end of the discussion to better describe work following on from the current study to better link this work to clinical applications & drug development.
Author Response
Reviewer #3:
The study provides a comprehensive investigation into the impact of metabolic stresses, specifically dyslipidemia and hyperglycemia, on OA pathogenesis. The focus on chondrocyte response and the potential protective role of EPA adds valuable insights to the understanding of OA at the molecular level.
We thank the reviewer for his/her kind words on our study and for his/her following comments and suggestions. We made our best to answer in the time allocated to respond.
The experimental design, involving the culture of rat chondrocytes with PA and/or EPA in normal or high glucose conditions, is well-structured. The use of qPCR to analyze gene expression and enzyme-linked immunosorbent assay to measure PGE2 release ensures a detailed examination of the molecular changes occurring in response to metabolic stresses. This reviewer finds it valuable to include other measures such as protein content (via Western blot or ELISA) and/or flow cytommetry to help support the findings throughout.
We acknowledge that WB for MMPs, collagen, and aggrecan and alternatively ELISA for MMPs could have be performed. However, as the most dramatic responses induced by PA and modulated by EPA were found to be inflammation, we deliberately chose to focus on PGE2 production to reflect both COX-2 and mPGES expression together with their activity. ELISA is the appropriate technique to detect PGE2 production in a quantitative manner, when WB for COX-2 and mPGES would only be qualitative and rather indicative of expression level but not bona fide activity level.
The qPCR readouts makes this reviewer wonder whether there were any changes to cell growth and would suggest the inclusion of some photographic images depicting each treatment condition to 'break up' the mRNA expression data presented & provide insight into whether cell growth/death could potentially impact the cell yield for qPCR.
We thank the reviewer for this suggestion. We had indeed originally performed some preliminary and non-exhaustive experiments to address a potential modulation of cell growth or death in our conditions and we did not find any obvious differences between treatments. That said, even if it is true that the initial amount of nucleic acid impact the accuracy and the reliability of data from qPCR experiments, normalization with a reference gene is useful to correct for differential cell yield/density. As a matter of fact, as mentioned in paragraph 4.3 in the materials and methods section, all our qPCR analysis for the expression level of each target gene was normalized to that of the housekeeping gene coding the ribosomal protein 29 (RPS29).
Another suggestion following on from these experiments would be to use an alternative readout via ddPCR to provide an absolute measure of copies/droplet (ng or RNA).
As far as we know, the very recent ddPCR technology is recommended for samples with low levels of nucleic acids (Cq>29) and/or with variable amounts of chemical and protein contaminants (Taylor S.C., Laperriere G., Germain H. Sci Rep, 2017, 7:2409). In our study, we analyzed the expression of target genes known to be fairly expressed in chondrocytes. Indeed, all our qPCR experiments showed raw data with Cq<27 and very low or no levels of background contaminants were found on the basis of spectrophotometric analysis. Therefore, all things considered, even if we had access to this top-notch technology, we believe ddPCR is not required in our study.
The findings regarding the up-regulation of MMP-3 and -13 by PA, along with the induction of COX-2 and mPGES, shed light on the inflammatory and degenerative processes associated with dyslipidemia and hyperglycemia. The exacerbating effect of high glucose concentration on chondrocyte response to PA underscores the potential synergy between these metabolic stresses in contributing to OA. It would be interesting to replicate these experiments in human cultured chondrocytes (a more relevant clinical model) to see if the data can be replicated or subject to interspecies variability.
We agree that experiments with human chondrocytes would be interesting and maybe more relevant to determine the effects of both dyslipidemia and hyperglycemia on cartilage cells. In the past, we ourselves have published numerous studies using either rat or human chondrocytes and we can strongly state, substantiated by a regimen of unpublished data, that there definitely exist interspecies variability. The most obvious reason for this variability is that rat chondrocytes are harvested from HEALTHY joints, when human chondrocytes are isolated from cartilage of OA (NON-HEALTHY) joints. These human chondrocytes may therefore originate from OA patients with pre-existing metabolic disorder that may result in impaired cell functions and thus may alter articular cell response. The high inter-individual variability due to different metabolic status will likely make the difference between treated and untreated groups non-significant unless the number of patients is very high. Overall, since our study, as stated in the title of our manuscript, concentrated on HEALTHY chondrocytes, replicating our data in human samples from various disease states and origins will not be appropriate, unless we had access to very rare and difficult to obtain chondrocytes from healthy dead donors.
The protective effects of EPA are a significant highlight of the study. The suppression of inflammatory effects induced by PA and glucose, coupled with a strong reduction in MMP-13 expression, suggests a potential therapeutic role for n-3 polyunsaturated fatty acids in preserving chondrocyte functions and maintaining cartilage homeostasis.
The identification of FFAR4 and FFAR1, in mediating EPA effects adds depth to the mechanistic understanding of the observed responses. The clarification that dyslipidemia associated with hyperglycemia may contribute to OA pathogenesis provides a valuable link between metabolic syndrome components and joint health. A minor suggestion would be to include a mechanism diagram for figure 6 to summarize the antagonistic roles & influences on biomarkers measured throughout this study.
We thank the reviewer for acknowledging our efforts to better understand the mechanisms unravelling the EPA effects. Even though as highlighted in the discussion, our data would need further exploration to fully picture the specific role of each FFAR in the EPA vs PA effects, we now include an additional figure (Fig. 8) that would schematically summarize our results and current hypothesis.
The manuscript is well written with minor edits in the attached PDF file. I would suggest improving the references included in the background to better introduce the current study. This reviewer would also like a future implications and impactful segment towards the end of the discussion to better describe work following on from the current study to better link this work to clinical applications & drug development.
We have thoughtfully considered and addressed accordingly the reviewer’ suggestions above, which also correspond to some in the attached PDF file. Our point-by-point responses to the reviewers’ inquiries are given below and changes in the manuscript are highlighted with red color text. All page numbers mentioned refer to the revised marked version after modifications were applied. English editing and changes have been made in the text according to the suggestions.
Introduction.
As recommended by the reviewer, references have been added where required. We added brief info about the techniques used on l.101-102 and l.106. We also combined the sentences as requested l.104.
Results
The reviewer recommended if possible to represent the data in the figures as dot plots, which are sometimes preferred to regular histograms. Following this suggestion, we modified the graphs accordingly. However as exemplified with the modified Fig6 below, we strongly feel that the figures became very confusing and difficult to read without providing additional useful information. For the sake of greater legibility, we would prefer to keep the original version of the figures. However, if the reviewer still wants us to modify them, we are ready to change all figures in the final and accepted version of the manuscript.
As we have defined the statistical tests used in the Mat&Met section, we feel it is unnecessary to mention it in each and every legends of the figures.
We acknowledge that « Hyperglycemia » is not appropriate for in vitro work. We have made changes accordingly in the legend of figure 3 l.162.
Data from qPCR experiments are expressed as fold change compared to the untreated control group, but statistical analysis have been actually performed on raw values.
Discussion
All the modifications requested by the reviewer have been made. The reviewer will find them highlighted in red in the marked version on l.326-331; l.420-426. We also included an additional Figure 8 (see also above).
